# Learning Concave Conditional Likelihood Models for Improved Analysis of Tandem Mass Spectra

**John T. Halloran**
Department of Public Health Sciences
University of California, Davis
jthalloran@ucdavis.edu

**David M. Rocke**
Department of Public Health Sciences
University of California, Davis
dmrocke@ucdavis.edu

## Abstract

The most widely used technology to identify the proteins present in a complex biological sample is tandem mass spectrometry, which quickly produces a large collection of spectra representative of the *peptides* (i.e., protein subsequences) present in the original sample. In this work, we greatly expand the parameter learning capabilities of a dynamic Bayesian network (DBN) peptide-scoring algorithm, Didea [25], by deriving emission distributions for which its conditional log-likelihood scoring function remains concave. We show that this class of emission distributions, called *Convex Virtual Emissions* (CVEs), naturally generalizes the log-sum-exp function while rendering both maximum likelihood estimation and conditional maximum likelihood estimation concave for a wide range of Bayesian networks. Utilizing CVEs in Didea allows efficient learning of a large number of parameters while ensuring global convergence, in stark contrast to Didea's previous parameter learning framework (which could only learn a single parameter using a costly grid search) and other trainable models [12, 13, 14] (which only ensure convergence to local optima). The newly trained scoring function substantially outperforms the state-of-the-art in both scoring function accuracy and downstream Fisher kernel analysis. Furthermore, we significantly improve Didea's runtime performance through successive optimizations to its message passing schedule and derive explicit connections between Didea's new concave score and related MS/MS scoring functions.

## 1 Introduction

A fundamental task in medicine and biology is identifying the proteins present in a complex biological sample, such as a drop of blood. The most widely used technology to accomplish this task is *tandem mass spectrometry* (*MS/MS*), which quickly produces a large collection of spectra representative of the peptides (i.e., protein subsequences) present in the original sample. A critical problem in MS/MS analysis, then, is the accurate identification of the peptide generating each observed spectrum.

The most accurate methods which solve this problem search a database of peptides derived from the mapped organism of interest. Such database-search algorithms score peptides from the database and return the top-ranking peptide per spectrum. The pair consisting of an observed spectrum and scored peptide are typically referred to as a *peptide-spectrum match* (*PSM*). Many scoring functions have been proposed, ranging from simple dot-products [2, 28], to cross-correlation based [8], to $p$-value based [10, 19, 16]. Recently, dynamic Bayesian networks (DBNs) have been shown to achieve the state-of-the-art in both PSM identification accuracy and post-search discriminative analysis, owing to their temporal modeling capabilities, parameter learning capabilities, and generative nature.

The first such DBN-based scoring function, Didea [25], used sum-product inference to efficiently compute a log-posterior score for highly accurate PSM identification. However, Didea utilized a

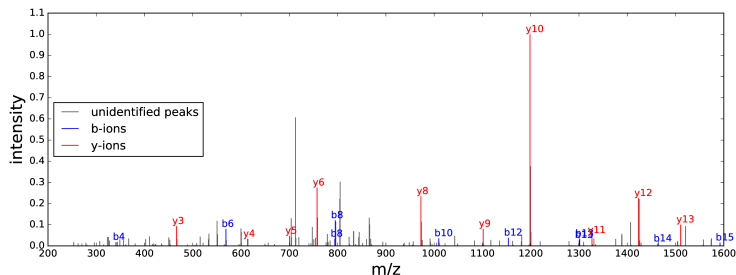

Figure 1: Example tandem mass spectrum with precursor charge $c^s = 2$ and generating peptide $x =$ TGPSPQPESQGSFYQR. Plotted in red and blue are, respectively, b- and y-ion peaks (discussed in Section 2.1), while unidentified peaks are colored gray.

complicated emission distribution for which only a single parameter could be learned through a costly grid search. Subsequently, a DBN for Rapid Identification of Peptides (DRIP) [13, 11], was shown in [12] to outperform Didea due to its ability to generatively learn a large number of model parameters. Most recently, DRIP's generative nature was further exploited to derive log-likelihood gradients detailing the manner in which peptides align with observed spectra [14]. Combining these gradients with a discriminative postprocessor [17], the resulting DRIP Fisher kernel substantially improved upon all state-of-the-art methods for downstream analysis on a large number of datasets.

However, while DRIP significantly improves several facets of MS/MS analysis due to its parameter learning capabilities, these improvements come at high runtime cost. In practice, DRIP inference is slow due to its large model complexity (the state-space grows exponentially in the lengths of both the observed spectrum and peptide). For instance, DRIP search required an order of magnitude longer than the slowest implementation of Didea for the timing tests in Section 5.2. Herein, we greatly improve upon all the analysis strengths provided by DRIP using the much faster Didea model. Furthermore, we optimize Didea's message passing schedule for a $64.2\%$ speed improvement, leading to runtimes two orders of magnitude faster than DRIP and comparable to less accurate(but widely used) methods. Thus, the work described herein not only improves upon state-of-the-art DBN analysis for effective parameter learning, scoring function accuracy, and downstream Fisher kernel recalibration, but also renders such analysis practical by significantly decreasing state-of-the-art DBN inference time.

In this work, we begin by discussing relevant MS/MS background and previous work. We then greatly expand the parameter learning capabilities of Didea by deriving a class of Bayesian network (BN) emission distributions for which both maximum likelihood learning and, most importantly, conditional maximum likelihood learning are concave. Called Convex Virtual Emissions (CVEs), we show that this class of emission distributions generalizes the widely used log-sum-exp function and naturally arises from the solution of a nonlinear differential equation representing convex conditions for general BN emissions. We incorporate CVEs into Didea to quickly and efficiently learn a substantial number of model parameters, considerably improving upon the previous learning framework. The newly trained model drastically improves PSM identification accuracy, outperforming all state-of-the-art methods over the presented datasets; at a strict FDR of $1\%$ and averaged over the presented datasets, the trained Didea scoring function identifies $16\%$ more spectra than DRIP and $17.4\%$ more spectra than the highly accurate and widely used MS-GF+ [19]. Under the newly parameterized model, we then derive a bound explicitly relating Didea's score to the popular XCorr scoring function, thus providing potential avenues to train XCorr using the presented parameter learning work.

With efficient parameter learning in place, we next utilize the new Didea model to improve MS/MS recalibration performance. We use gradient information derived from Didea's conditional log-likelihood in the feature-space of a kernel-based classifier [17]. Training the resulting conditional Fisher kernel substantially improves upon the state-of-the-art recalibration performance previously achieved by DRIP; at a strict FDR of $1\%$, discriminative recalibration using Didea's conditional Fisher kernel results in an average $11.3\%$ more identifications than using the DRIP Fisher kernel. Finally, we conclude with a discussion of several avenues for future work.

## 2 Tandem mass spectrometry

With a complex sample as input, a typical MS/MS experiment begins by cleaving the proteins of the sample into peptides using a digesting enzyme, such as trypsin. The digested peptides are then separated via liquid chromatography and undergo two rounds of mass spectrometry. The first round of mass spectrometry measures the mass and charge of the intact peptide, referred to as the *precursor mass* and *precursor charge*, respectively. Peptides are then fragmented into prefix and suffix ions. The mass-to-charge (*m/z*) ratios of the resulting fragment ions are measured in the second round of mass spectrometry, producing an observed spectrum of m/z versus intensity values representative of the fragmented peptide. The output of this overall process is a large collection of spectra (often numbering in the hundreds-of-thousands), each of which is representative of a peptide from the original complex sample and requires identification. The x-axis of such observed spectra denotes m/z, measured in thomsons (Th), and y-axis measures the intensity at a particular m/z value. A sample such observed spectrum is illustrated in Figure 1.

### 2.1 Database search and theoretical spectra

Let $s \in S$ be an observed spectrum with precursor m/z $m^s$ and precursor charge $c^s$, where $S$ is the universe of tandem mass spectra. The generating peptide of $s$ is identified by searching a database of peptides, as follows. Let $\mathbb{P}$ be the universe of all peptides and $x \in \mathbb{P}$ be an arbitrary peptide of length $l$. $x = x_1 \dots x_l$ is a string comprised of characters called *amino acids*, the dictionary size of which are 20. We denote peptide substrings as $x_{i:j} = x_i, \dots, x_j$, where $i > 0, j \leq l, i < j$, and the mass of $x$ as $m(x)$. Given a peptide database $\mathcal{D} \subseteq \mathbb{P}$, the set of peptides considered is constrained to those within a precursor mass tolerance window $w$ of $m^s$. The set of *candidate peptides* to be scored is thus $D(s, \mathcal{D}, w) = \{x : x \in \mathcal{D}, |\frac{m(x)}{c^s} - m^s| \leq w\}$. Using a scoring function $\psi : \mathbb{P} \times S \to \mathbb{R}$, a database search outputs the top-scoring PSM, $x^* = \text{argmax}_{x \in \mathcal{D}} \psi(x, s)$.

In order to score a PSM, the idealized fragment ions of $x$ are first collected into a theoretical spectrum. The most commonly encountered fragment ions are called *b-ions* and *y-ions*. B- and y-ions correspond to prefix and suffix mass pairs, respectively, such that the precursor charge $c^s$ is divided amongst the pair. For b-ion charge $c_b \leq c^s$, the $k$th b-ion and the accompanying y-ion are then, respectively,

$$b(m(x_{1:k}), c_b) = \frac{m(x_{1:k}) + c_b}{c_b} = \frac{\left[\sum_{i=1}^{k} m(x_i)\right] + c_b}{c_b}, \quad y(m(x_{k+1:l}), c_y) = \frac{\left[\sum_{i=k+1}^{l} m(x_i)\right] + 18 + c_y}{c_y},$$

where $c_y$ is the y-ion charge, the b-ion offset corresponds to a $c_b$ charged hydrogen atom, and the y-ion offset corresponds to a $c_y$ charged hydrogen atom plus a water molecule. For singly charged spectra $c^s = 1$, only singly charged fragment ions are detectable, so that $c_b = c_y = 1$. For higher precursor charge states $c^s \geq 2$, the total charge is split between each b- and y-ion pair, so that $0 < c_b < c^s$ and $c_y = c^s - c_b$. The annotated b- and y-ions of an identified observed spectrum are illustrated in Figure 1.

## 3 Previous work

Many database search scoring algorithms have been proposed, each of which is characterized by the scoring function they employ. These scoring functions have ranged from dot-products (X!Tandem [2] and Morpheus [28]), to cross-correlation based (XCorr [8]), to exact $p$-values computed over linear scores [19, 16]. Recently, DBNs have been used to substantially improve upon the accuracy of previous approaches.

In the first such DBN, Didea [25], the time series being modeled is the sequence of a peptide's amino acids (i.e., an amino acid is observed in each frame) and the quantized observed spectrum is observed in each frame. In successive frames, the sequence of b- and y-ions are computed and used as indices into the observed spectrum via virtual evidence [24]. A hidden variable in the first frame, corresponding to the amount to shift the observed spectrum by, is then marginalized in order to compute a conditional log-likelihood probability consisting of a foreground score minus a background score, similar in form to XCorr (described in Section 4.2.1). The resulting scoring function outperformed the most accurate scoring algorithms at the time (including MS-GF+, then called MS-GFDB) on a majority of datasets. However, parameter learning in the model was severely limited and inefficient; a single hyperparameter controlling the reweighting of peak intensities was learned via an expensive grid search, requiring repeated database searches over a dataset.

Subsequent work saw the introduction of DRIP [12, 13], a DBN with substantial parameter learning capabilities. In DRIP, the time series being modeled is the sequence of observed spectrum peaks (i.e., each frame in DRIP corresponds to an observed peak) and two types of prevalent phenomena are explicitly modeled via sequences of random variables: spurious observed peaks (called insertions) and absent theoretical peaks (called deletions). A large collection of Gaussians parameterizing the m/z axis are generatively learned, via expectation-maximization (EM) [4], and used to score observed peaks. DRIP then uses max-product inference to calculate the most probable sequences of insertions and deletions in order to score PSMs.

In practice, the majority of PSM scoring functions discussed are typically poorly calibrated, i.e., it is often difficult to compare the PSM scores across different spectra. In order to combat such poor calibration, postprocessors are commonly employed to recalibrate PSM scores [17, 26, 27]. In recent work, DRIP's generative framework was further exploited to calculate highly accurate features based on the log-likelihood gradients of its learnable parameters. Combining these new gradient-based features with a popular kernel-based classifier for recalibrating PSM scores [17], the resulting Fisher kernel was shown to significantly improve postprocessing accuracy [14].

# 4 Didea

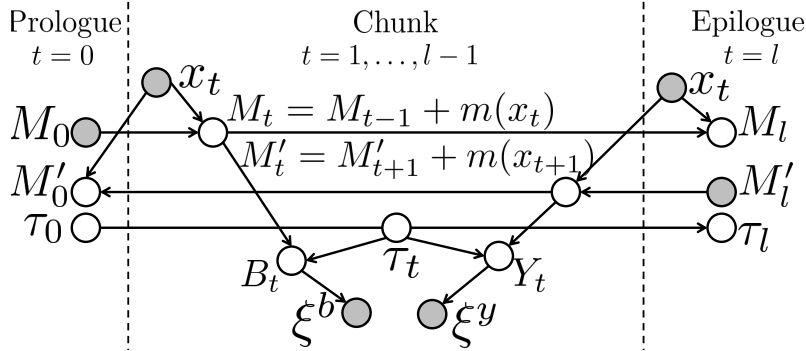

Figure 2: Graph of Didea. Unshaded nodes are hidden, shaded nodes are observed, and edges denote deterministic functions of parent variables.

We now derive Didea's scoring function in detail. The graph of Didea is displayed in Figure 2. Shaded variables are observed and unshaded variables are hidden (random). Groups of variables are collected into time instances called *frames*, where the first frame is called the prologue, the final frame is called the epilogue, and the *chunk* dynamically expands to fill all frames in between. Let $0 \leq t \leq l$ be an arbitrary frame. The amino acids of a peptide are observed in each frame after the prologue. The variable $M_t$ successively accumulates the prefix masses of the peptide such that $p(M_0 = 0) = 1$ and $p(M_t = M_{t-1} + m(x_t)|M_{t-1}, x_t) = 1$, while the variable $M'_t$ successively accumulates the suffix masses of the peptide such that $p(M'_l = 0) = 1$ and $p(M'_t = M'_{t+1} + m(x_{t+1})|M'_{t+1}, x_{t+1}) = 1$. Denoting the maximum spectra shift as $L$, the shift variable $\tau_0 \in [-L, L]$ is hidden, uniform, and deterministically copied by its descendents in successive frames, such that $p(\tau_t = \bar{\tau}|\tau_0 = \bar{\tau}) = 1$ for $t > 1$.

Let $s \in \mathbb{R}^{\bar{o}+1}$ be the binned observed spectrum, i.e., a vector of length $\bar{o} + 1$ whose $i$th element is $s(i)$, where $\bar{o}$ is the maximum observable discretized m/z value. Shifted versions of the $t$th b- and y-ion pair (where the shift is denoted by subscript) are deterministic functions of the shift variable as well as prefix and suffix masses, i.e., $p(B_t = b_{\tau_t}(M_t, 1)|M_t, \tau_t) = p(B_t = \max(\min(b(M_t, 1) - \tau_t, 0), \bar{o})|M_t, \tau_t), p(Y_t = y_{\tau_t}(m(x_{t+1:l}), 1)|M'_t, \tau_t) = p(Y_t = \max(\min(y(m(x_{t+1:l}), 1) - \tau, 0), \bar{o})|M'_t, \tau_t)$, respectively. $\xi^b$ and $\xi^y$ are *virtual evidence children*, i.e., leaf nodes whose conditional distribution need not be normalized (only non-negative) to compute posterior probabilities in the DBN. A comprehensive overview of virtual evidence is available in [15]. $\xi^b$ and $\xi^y$ compare the b- and y-ions, respectively, to the observed spectrum, such that $p(\xi^b|B_t) = f(s(b_{\tau_t}(m(x_{1:t}), 1))), p(\xi^y|Y_t) = f(s(y_{\tau_t}(m(x_{t+1:l}), 1)))$, where $f$ is a non-negative emission function.

Let $\mathbf{1}_{\{\cdot\}}$ denote the indicator function. Didea's log-likelihood is then $\log \mathrm{p}(\tau_0 = \bar{\tau}, x, s)$

$= \log p(\tau_0 = \bar{\tau})p(M_0)p(M_0'|M_1', x_1)p(M_l')p(M_l|M_{l-1}, x_1)+$

$\qquad \log \prod_{t=1}^{l} [p(\tau_t|\tau_{t-1})p(M_t|M_{t-1}, x_t)p(M_t'|M_{t+1}', x_{t+1})p(B_t|M_t, \tau_t)p(Y_t|M_t', \tau_t)p(\xi^b|B_t)p(\xi^y|Y_t)]$

$= \log p(\tau_0 = \bar{\tau}) + \log \prod_{t=1}^{l-1} \big(\mathbf{1}_{\{\tau_t = \bar{\tau} \wedge M_t = m(x_{1:t}) \wedge M_t' = m(x_{t+1:l})\}} p(B_t|M_t, \tau_t)p(Y_t|M_t', \tau_t)p(\xi^b|B_t)p(\xi^y|Y_t)\big)$

$= \log p(\tau_0 = \bar{\tau}) + \log \prod_{t=1}^{l-1} p(\xi^b|b_{\bar{\tau}}(M_t, 1))p(\xi^y|y_{\bar{\tau}}(M_t', 1))]$

$= \log p(\tau_0 = \bar{\tau}) + \sum_{t=1}^{l-1} \big(\log f(s_{\bar{\tau}}(b(m(x_{1:t}), 1))) + \log f(s_{\bar{\tau}}(y(m(x_{t+1:l}), 1)))\big).$

In order to score PSMs, Didea computes the conditional log-likelihood

$$\psi(s, x) = \log \mathrm{p}(\tau_0 = 0|x, s) = \log \mathrm{p}(\tau_0 = 0, x, s) - \log \sum_{\bar{\tau}=-L}^{L} \mathrm{p}(\tau_0 = \bar{\tau})\mathrm{p}(x, s_{\bar{\tau}}|\tau_0 = \bar{\tau})$$

$$= \log \mathrm{p}(\tau_0 = 0, x, s) - \log \frac{1}{|\tau_0|} \sum_{\bar{\tau}=-L}^{L} \mathrm{p}(x, s_\tau|\tau_0 = \bar{\tau}). \tag{1}$$

As previously mentioned, $\psi(s, x)$ is a foreground score minus a background score, where the background score consists of averaging over $|\tau_0|$ shifted versions of the foreground score, much like the XCorr scoring function. Thus, Didea may be thought of as a probabilistic analogue of XCorr.

### 4.1 Convex Virtual Emissions for Bayesian networks

Consider an arbitrary Bayesian network where the observed variables are leaf nodes, as is common in a large number of time-series models such as hidden Markov models (HMMs), hierarchical HMMs [22], DBNs for speech recognition [5], hybrid HMMs/DBNs [3], as well as DRIP and Didea. Let $E$ be the set of observed random variables, $H$ be the hypothesis space composed of the cross-product of the $n$ hidden discrete random variables in the network, and $h \in H$ be an arbitrary *hypothesis* (i.e., an instantiation of the hidden variables). As is the case in Didea, often desired is the log-posterior probability $\log p(h|E) = \log \frac{p(h,E)}{p(E)} = \log \frac{p(h,E)}{\sum_{\bar{h} \in H} p(\bar{h}, E)} = \log p(h, E) - \log \sum_{h \in H} p(h)p(E|h)$. Under general assumptions, we'd like to find emission functions for which $\log p(h|E)$ is concave.

Assume $p(h)$ and $p(E|h)$ are non-negative, that the emission density $p(E|h)$ is parameterized by $\theta$ (which we'd like to learn), and that there is a parameter $\theta_h$ to be learned for every hypothesis of latent variables (though if we have fewer parameters, parameter estimation becomes strictly easier). We make this parameterization explicit by denoting the emission distributions of interest as $p_{\theta_h}(E|h)$. Assume that $p_{\theta_h}(E|h)$ is smooth on $\mathbb{R}$ for all $h \in H$. Applying virtual evidence for such models, $p_{\theta_h}(E|h)$ need not be normalized for posterior inference (as well as Viterbi inference and comparative inference between sets of observations; an extensive review of virtual evidence may be found in [15]).

Given the factorization of the joint distribution described by the BN, the quantity $p_{\theta_h}(h, E) = p(h)p_{\theta_h}(E|h)$ may often be efficiently computed for any given $h$. Thus, the computationally difficult portion of $\log p(h|E)$ is the calculation of the log-likelihood in the denominator, wherein all hidden variables are marginalized over. We therefore first seek emission functions for which the log-likelihood $\log p(E) = \log \sum_{h \in H} p(h)p_{\theta_h}(E|h)$ is convex. For such emission functions, we have the following theorem.

**Theorem 1.** *The unique convex functions of the form $\log \sum_{h \in H} p(h)p_{\theta_h}(E|h)$, such that $(p'_{\theta_h}(E|h))^2 - p''_{\theta_h}(E|h)p_{\theta_h}(E|h) = 0$, are $\log \sum_{h \in H} p(h)p_{\theta_h}(E|h) = \log \sum_{h \in H} \alpha_h e^{\beta_h \theta_h}$, where $\alpha_h = p(h)a_h$ and $a_h, \beta_h$ are hyperparameters.*

The full proof of Theorem 1 is given in [15]. The nonlinear differential equation $(p'_{\theta_h}(E|h))^2 - p''_{\theta_h}(E|h)p_{\theta_h}(E|h) = 0$ describes the curvature of the desired emission functions and arises from the necessary and sufficient conditions for twice differentiable convex functions (i.e., the Hessian

must be p.s.d.) and the Cauchy-Schwarz inequality. Particular values of the hyperparameters $a_h$ and $\beta_h$ correspond to unique initial conditions for this nonlinear differential equation. Note that when $\alpha_h = 1, p_{\theta_h}(E|h) = e^{\theta_h}$, we have the well-known log-sum-exp (LSE) convex function. Thus, this result generalizes the LSE function to a broader class of convex functions.

We call the unique class of convex functions which arise from solving the nonlinear differential in Theorem 1, $p_{\theta_h}(E|h) = a_h e^{\beta_h \theta_h}$, Convex Virtual Emissions (CVEs). Note that utilizing CVEs, maximimum likelihood estimation (i.e., $\mathrm{argmax}_\theta - \log \sum_{h \in H} p(h) p_{\theta_h}(E|h)$) is thus concave and guaranteed to converge to a global optimum. Furthermore, and most importantly for Didea, we have the following result for the conditional log-likelihood (the full proof of which is in [15]).

**Corollary 1.1.** *For convex* $\log p(E) = \log \sum_{h \in H} p(h) p_{\theta_h}(E|h)$ *such that* $(p'_{\theta_h}(E|h))^2 - p''_{\theta_h}(E|h)p_{\theta_h}(E|h) = 0$, *the log-posterior* $\log p_\theta(h|E)$ *is concave in* $\theta$.

Thus, utilizing CVEs, conditional maximum likelihood estimation is also rendered concave.

## 4.2 CVEs in Didea

In [25], the virtual evidence emission function to score peak intensities was $f_\lambda(s(i)) = 1 - \lambda e^{-\lambda} + \lambda e^{-\lambda(1-s(i))}$. Under this function, Didea was shown to perform well on a variety of datasets. However, this function is non-convex and does not permit efficient parameter learning; although only a single model parameter, $\lambda$, was trained, learning required a grid search wherein each step consisted of a database search over a dataset and subsequent target-decoy analysis to assess each new parameter value. While this training scheme is already costly and impractical, it quickly becomes infeasible when looking to learn more than a single model parameter.

We use CVEs to render Didea's conditional log-likelihood concave given a large number of parameters. To efficiently learn a distinct observation weight $\theta_\tau$ for each spectral shift $\tau \in [-L, L]$, we thus utilize the emission function $f_{\theta_\tau}(s(i)) = e^{\theta_\tau s(i)}$. Denote the set of parameters per spectra shift as $\theta = \{\theta_{-L}, \ldots, \theta_L\}$. Due to the concavity of Equation 1 using $f_{\theta_\tau}$ under Corollary 1.1, given $n$ PSM training pairs $(s^1, x^1), \ldots, (s^n, x^n)$, the learned parameters $\theta^* = \mathrm{argmax}_\theta \sum_{i=1}^n \psi_\theta(s^i, x^i)$ are guaranteed to converge to a global optimum. Further analysis of Didea's scoring function under this new emission function may be found in [15], including the derivation of the new model's gradients (i.e., conditional Fisher scores).

### 4.2.1 Relating Didea's conditional log-likelihood to XCorr using CVEs

XCorr [8], the very first database search scoring function for peptide identification, remains one of the most widely used tools in the field today. Owing to its prominence, XCorr remains an active subject of analysis and continuous development [20, 23, 7, 6, 16, 21, 9, 14]. As previously noted, the scoring functions of XCorr and Didea share several similarities in form, where, in fact, the former served as the motivating example in [25] for both the design of the Didea model and its posterior-based scoring function. While cosmetic similarities have thus far been noted, the reparameterization of Didea's conditional log-likelihood using CVEs permits the derivation of an explicit relationship between the two.

Let $u$ be the theoretical spectrum of peptide $x$. As with Didea, let $L$ be the maximum spectra shift considered and, for shift $\tau$, denote a vector shift via subscript, such that $s_\tau$ is the vector of observed spectrum elements shifted by $\tau$ units. In order to compare $u$ and $s$, XCorr is thus computed as $\mathrm{XCorr}(s, x) = u^T s - \frac{1}{2L+1} \sum_{\tau=-L}^L u^T s_\tau$. Intuitively, the cross-correlation background term is meant to penalize overfitting of the theoretical spectrum. Under the newly parameterized Didea conditional log-likelihood described herein, we have the following theorem explicitly relating the XCorr and Didea scoring functions.

**Theorem 2.** *Assume the PSM scoring function* $\psi(s, x)$ *is that of Didea (i.e., Equation 1) where the emission function* $f_{\theta_\tau}(s(i))$ *has uniform weights* $\theta_i = \theta_j$, *for* $i, j \in [-L, L]$. *Then* $\psi(s, x) \leq \mathcal{O}(XCorr(s, x))$.

The full proof of Theorem 2 may be found in [15]. Thus, Didea's scoring function effectively serves to lower bound XCorr. This opens possible avenues for extending the learning results detailed herein to the widely used XCorr function. For instance, a natural extension is to use a variational Bayesian

inference approach and learn XCorr parameters through iterative maximization of the Didea lower bound, made efficient by the concavity of new Didea model derived in this work.

## 4.3  Faster Didea sum-product inference

We successively improved Didea's inference time when conducting a database search using the intensive charge-varying model (discussed in [15]). Firstly, we removed the need for a backward pass by keeping track of the foreground log-likelihood during the forward pass (which computes the background score, i.e., the probability of evidence in the model). Next, by exploiting the symmetry of the spectral shifts, we cut the effective cardinality of $\tau$ in half during inference. While this requires twice as much memory in practice, this is not close to being prohibitive on modern machines. Finally, a large portion of the speedup was achieved by offsetting the virtual evidence vector by $|\tau|$ and pre/post buffering with zeros and offsetting each computed b- and y-ion by $|\tau|$. Under this construction, the scores do not change, but, during inference, we are able to shift each computed b- and y-ion by $\pm\tau$ without requiring any bound checking. Hashing virtual evidence bin values by b-/y-ion value and $\tau$ was also pursued, but did not offer any runtime benefit over the aforementioned speedups (due to the cost of constructing the hash table per spectrum).

## 5  Results

In practice, assessing peptide identification accuracy is made difficult by the lack of ground-truth encountered in real-world data. Thus, it is most common to estimate the *false discovery rate* (*FDR*) [1] by searching a decoy database of peptides which are unlikely to occur in nature, typically generated by shuffling entries in the target database [18]. For a particular score threshold, $t$, the FDR is calculated as the proportion of decoys scoring better than $t$ to the number of targets scoring better than $t$. Once the target and decoy PSM scores are calculated, a curve displaying the FDR threshold versus the number of correctly identified targets at each given threshold may be calculated. In place of FDR along the x-axis, we use the *q-value* [18], defined to be the minimum FDR threshold at which a given score is deemed to be significant. As many applications require a search algorithm perform well at low thresholds, we only plot $q \in [0, 0.1]$.

The benchmark datasets and search settings used to recently evaluate the DRIP Fisher kernel in [14] are adapted in this work. The charge-varying Didea model (which integrates over multiple charge states, further described in [15]) with concave emissions (described in Section 4.2) was used to score and rank database peptides. Concave Didea parameters were learned using the high-quality PSMs used to generatively train the DRIP model in [13] and gradient ascent. Didea's newly trained database-search scoring function is benchmarked against the Didea model from [25] trained using a costly grid search for a single parameter (denoted as "Didea-0") and four other state-of-the-art scoring algorithms: DRIP, MS-GF+, XCorr $p$-values, and XCorr.

DRIP searches were conducted using the DRIP Toolkit and the generatively trained parameters described in [12, 13]. MS-GF+, one of the most accurate search algorithms in wide-spread use, was run using version 9980, with PSMs ranked by E-value. XCorr and XCorr $p$-value scores were collected using Crux v2.1.17060. All database searches were run using a $\pm3.0$Th mass tolerance, XCorr flanking peaks not allowed in Crux searches, and all search algorithm settings otherwise left to their defaults. Peptides were derived from the protein databases using trypsin cleavage rules without suppression of proline and a single fixed carbamidomethyl modification was included.

The resulting database-search accuracy plots are displayed in Figure 3. The trained Didea model outperforms all competitors across all presented datasets; compared to highly accurate scoring algorithms DRIP and MS-GF+, the trained Didea scoring function identifies 16% more spectra than DRIP and 17.4% more spectra than MS-GF+, at a strict FDR of 1% averaged over the presented datasets. This high-level performance is attributable to the expanded and efficient parameter learning framework, which greatly improves upon the limited parameter learning capabilities of the original Didea model, identifying 9.8% more spectra than Didea-0 at a strict FDR of 1% averaged over the presented datasets.

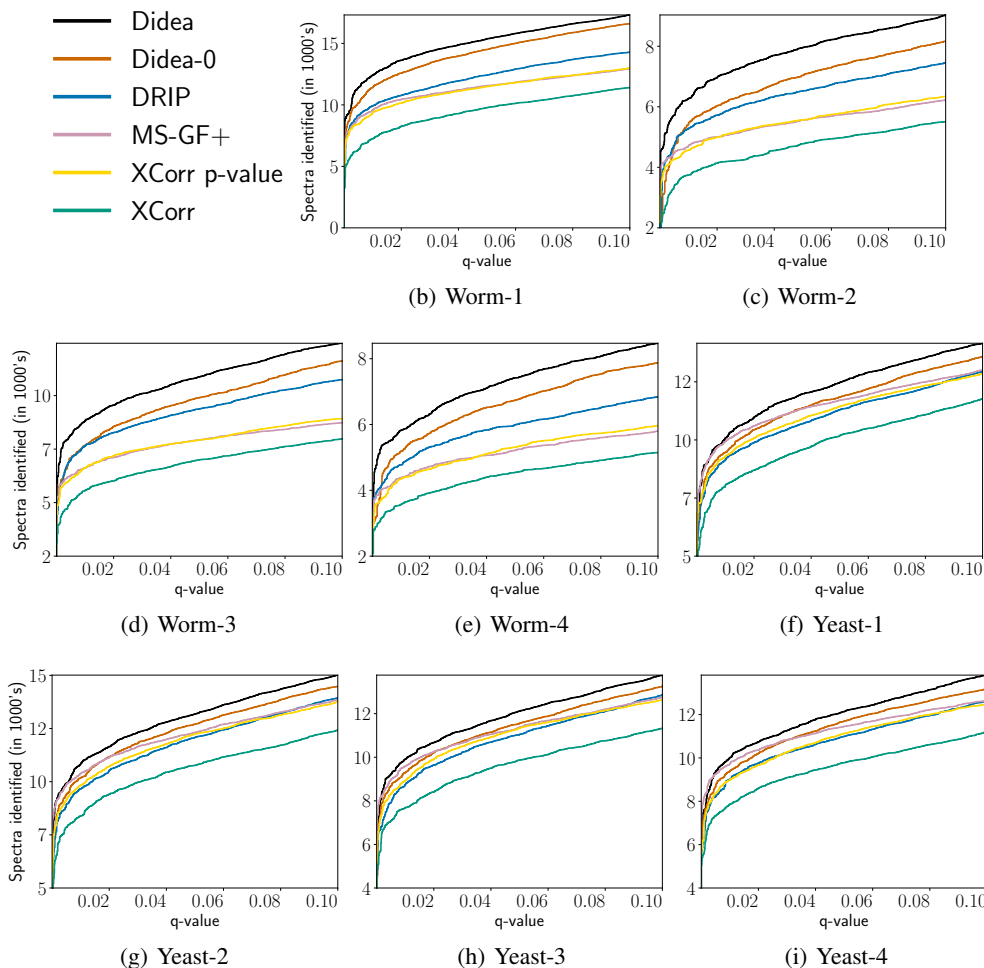

Figure 3: Database search accuracy plots measured by $q$-value versus number of spectra identified for worm (*C. elegans*) and yeast (*Saccharomyces cerevisiae*) datasets. All methods are run with as equivalent settings as possible. The Didea charge-varying model (described in [15]) was used to score PSMs, with "Didea" denoting the model trained per charge state using the concave framework described in Section 4.2 and "Didea-0" denoting the model from [25] trained using a grid search. DRIP, another DBN-based scoring function, was run using the generatively learned parameters described in [13].

## 5.1 Conditional Fisher kernel for improved discriminative analysis

Facilitated by Didea's effective parameter learning framework, we look to leverage gradient-based PSM information to aid in discriminative postprocessing analysis. We utilize the same set of features as the DRIP Fisher kernel [14]. However, in order to measure the relative utility of the gradients under study, we replace the DRIP log-likelihood gradients with Didea gradient information (derived in [15]). These features are used to train an SVM classifier, Percolator [17], which recalibrates PSM scores based on the learned decision boundary between input targets and decoys. Didea's resulting conditional Fisher kernel is benchmarked against the DRIP Fisher kernel and the previously benchmarked scoring algorithms using their respective standard Percolator features sets.

DRIP Kernel features were computed using the customized version of the DRIP Toolkit from [14]. MS-GF+ Percolator features were collected using `msgf2pin` and XCorr/XCorr $p$-value features collected using Crux. For the resulting postprocessing results, the trained Didea scoring function outperforms all competitors, identifying 12.3% more spectra than DRIP and 13.4% more spectra than MS-GF+ at a strict FDR of 1% and averaged over the presented datasets. The full panel of results is displayed in [15]. Compared to DRIP's log-likelihood gradient features, the conditional

log-likelihood gradients of Didea contain much richer PSM information, thus allowing Percolator to better distinguish target from decoy PSMs for much greater recalibration performance.

## 5.2 Optimized exact sum-product inference for improved Didea runtime

Implementing the speedups to exact Didea sum-product inference described in Section 4.3, we benchmark the optimized search algorithm using $1,000$ randomly sampled spectra (with charges varying from 1+ to 3+) from the Worm-1 dataset and averaged database-search times (reported in wall clock time) over 10 runs. The resulting runtimes are listed in Table 1. DRIP was run using the DRIP Toolkit and XCorr $p$-values were collected using Crux v2.1.17060. All benchmarked search algorithms were run on the same machine with an Intel Xeon E5-2620 and 64GB RAM. The described optimizations result in a $64.2\%$ runtime improvement, and brings search time closer to less accurate, but faster, search algorithms.

|         | Didea-0 | Didea Opt. | XCorr $p$-values | DRIP     |
|---------|---------|------------|------------------|----------|
| runtime | 19.1175 | 6.8535     | 2.2955           | 143.4712 |

Table 1: Database search runtimes per spectrum, in seconds, searching 1,000 worm spectra randomly sampled from the Worm-1 dataset. "Didea-0" is the implementation of Didea used in [25] and "Didea Opt" is the speed-optimized implementation described herein. All reported search algorithm runtimes were averaged over 10 runs.

## 6 Conclusions and future work

In this work, we've derived a widely applicable class of Bayesian network emission distributions, CVEs, which naturally generalize the convex log-sum-exp function and carry important theoretical properties for parameter learning. Using CVEs, we've substantially improved the parameter learning capabilities of the DBN scoring algorithm, Didea, by rendering its conditional log-likelihood concave with respect to a large set of learnable parameters. Unlike previous DBN parameter learning solutions, which only guarantee convergence to a local optimum, the new learning framework thus guarantees global convergence. Didea's newly trained database-search scoring function significantly outperforms all state-of-the-art scoring algorithms on the presented datasets. With efficient parameter learning in hand, we derived the gradients of Didea's conditional log-likelihood and used this gradient information in the feature space of a kernel-based discriminative postprocessor. The resulting conditional Fisher kernel once again outperforms the state-of-the-art on all presented datasets, including a highly accurate, recently proposed Fisher kernel. Furthermore, we successively optimized Didea's message passing schedule, leading to DBN analysis times two orders of magnitude faster than other leading DBN tools for MS/MS analysis. Thus, the presented results improve upon all aspects of state-of-the-art DBN analysis for MS/MS. Finally, using the new learning framework, we've proven that Didea is proportionally lower bounds the widely used XCorr scoring function.

There are a number of exciting avenues for future work. Considering the large amount of PSM information held in the gradient space of Didea's conditional log-likelihood, we plan on pursuing kernel-based approaches to peptide identification using the Hessian of the scoring function. This is especially exciting given the high degree of recalibration accuracy provided by Percolator, a kernel-based post-processor. Using a variational approach, we also plan on investigating parameter learning options for XCorr given the Didea lower bound and the concavity of Didea's parameterized scoring function. Finally, in perhaps the most ambitious plan for future work, we plan to further build upon Didea's parameter learning framework by learning distance matrices between the theoretical and observed spectra. Such matrices naturally generalize the class of CVEs derived herein.

**Acknowledgments**: This work was supported by the National Center for Advancing Translational Sciences (NCATS), National Institutes of Health, through grant UL1 TR001860.

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
