[Supplementary Material · supplementary.pdf]

# Learning Concave Conditional Likelihood Models for Improved Analysis of Tandem Mass Spectra: Supplementary Material

**John T. Halloran**
Department of Public Health Sciences
University of California, Davis
jthalloran@ucdavis.edu

**David M. Rocke**
Department of Public Health Sciences
University of California, Davis
dmrocke@ucdavis.edu

## 1 Virtual evidence

*Virtual evidence* [8] is a versatile mechanism by which, given a Bayesian network (BN), we may model the conditional distributions of observed leaf nodes using nonnegative distributions which need not be normalized for a great deal of probabilistic quantities commonly of interest. Formally, for a BN $G = (V, E)$ with edge set $E$ and vertex set $V$, let $O \subseteq V = \{o_1, \ldots, o_n\}$ be a set of observed leaf nodes, and $H = V \setminus O$. For any $S \subseteq V$, denote the set of parents for all variables in $S$ as $\pi_S$. The conditional distribution of $o_i \in O$ is then $\mathrm{p}(o_i | \pi_{o_i}) = \frac{\psi_i(o_i, \pi_{o_i})}{Z_i}$, where $\psi_i : \mathbb{R}^{|\pi_{o_i}|+1} \to \mathbb{R}_+$, $Z_i$ is a normalizing constant, and $\mathbb{R}_+$ is the set of nonnegative reals. The joint distribution over $G$ is thus

$$
\begin{aligned}
\mathrm{p}(O, H) &= \mathrm{p}(O, \pi_O)\mathrm{p}(H | O, \pi_O) \\
&= \mathrm{p}(H | O, \pi_O) \prod_{i=1}^{n} \mathrm{p}(o_i | \pi_{o_i}) \\
&= \mathrm{p}(H | O, \pi_O) \prod_{i=1}^{n} \frac{\psi_i(o_i, \pi_{o_i})}{Z_i} \\
&= \frac{1}{Z} \mathrm{p}(H | O, \pi_O) \prod_{i=1}^{n} \psi_i(o_i, \pi_{o_i}),
\end{aligned}
\tag{1}
$$

where $Z = \prod_{i=1}^{n} Z_i$. Note that $\mathrm{p}(H | O, \pi_O)$ may further factorize depending on the BN.

We now show that, for many practical applications of interest, we do not need the normalization constants $Z_1, \ldots, Z_n$ when computing the Viterbi path, the probability of evidence, or posteriors of $G$. We define the following quantity, which will prove useful throughout our discussion,

$$
\mathrm{p}'(O, H) = Z\mathrm{p}(O, H) = \mathrm{p}(H | O, \pi_O) \prod_{i=1}^{n} \psi_i(o_i, \pi_{o_i}).
$$

## 1.1 Posterior probabilities

Consider the case where we are interested in posterior probabilities in $G$, i.e., for $H' \subseteq H$, we would like to compute $p(H'|O)$. We thus have

$$
\begin{aligned}
p(H'|O) &= \frac{p(H',O)}{p(O)} \\
&= \frac{\sum_{X \in H \setminus H'} \frac{1}{Z} p(H|O, \pi_O) \prod_{i=1}^{n} \psi_i(o_i, \pi_{o_i})}{\sum_{X \in H} \frac{1}{Z} p(H|O, \pi_O) \prod_{i=1}^{n} \psi_i(o_i, \pi_{o_i})} \\
&= \frac{\sum_{X \in H \setminus H'} p(H|O, \pi_O) \prod_{i=1}^{n} \psi_i(o_i, \pi_{o_i})}{\sum_{X \in H} p(H|O, \pi_O) \prod_{i=1}^{n} \psi_i(o_i, \pi_{o_i})} \\
&= \frac{\sum_{X \in H \setminus H'} p'(O, H)}{\sum_{X \in H} p'(O, H)}.
\end{aligned}
$$

Thus, the normalization constants are not necessary to compute posteriors over the hidden variables.

## 1.2 Probability of evidence and Viterbi score

The *probability of evidence*, or *score*, is the quantity computed after summing over all hidden variables in $G$. It is often used to score different sets of observations. For instance, consider that we have $m$ sets of observations $O^i = \{o_1^i, \ldots, o_n^i\}$ for $i = 1, \ldots, m$, where each of the sets may contain observed information regarding a peptide, observed spectrum, or both. Now, consider we'd like to score and rank each $O^i$, using $G$, as $p(O^i)$ in order to choose the maximum scoring set of observations $O^* = \operatorname{argmax}_{O^i, i=1,\ldots,m} p(O^i)$. This is the general scenario we have when performing an MS/MS database search. From Equation 1, we thus have

$$
\begin{aligned}
O^* &= \operatorname*{argmax}_{O^i, i=1,\ldots,m} \; p(O^i) \\
&= \operatorname*{argmax}_{O^i, i=1,\ldots,m} \sum_{X \in H} \frac{1}{Z} p(H|O^i, \pi_O) \prod_{j=1}^{n} \psi_j(o_j, \pi_{o_j}) \\
&= \operatorname*{argmax}_{O^i, i=1,\ldots,m} \frac{1}{Z} \sum_{X \in H} p(H|O^i, \pi_{O^i}) \prod_{j=1}^{n} \psi_j(o_j, \pi_{o_j}) \\
&= \operatorname*{argmax}_{O^i, i=1,\ldots,m} \sum_{X \in H} p'(O, H).
\end{aligned}
$$

This is the case since $p(O^i, H) \propto p'(O^i, H)$ and, since $Z$ is constant with respect to the sum, $\sum_{X \in H} p'(O, H) \propto \sum_{X \in H} p(O, H)$. Thus, we need not worry about the normalizing constants $Z_1, \ldots, Z_n$ if we need only make decisions based on the relative scores between sets of observations.

Now, instead of the probability of evidence, consider we would like to score each $O^i$ with their Viterbi score. We thus have

$$
\begin{aligned}
O^* &= \operatorname*{argmax}_{O^i, i=1,\ldots,m} \max_{X \in H} \frac{1}{Z} p(H|O^i, \pi_{O^i}) \prod_{j=1}^{n} \psi_j(o_j, \pi_{o_j}) \\
&= \operatorname*{argmax}_{O^i, i=1,\ldots,m} \frac{1}{Z} \max_{X \in H} p(H|O^i, \pi_{O^i}) \prod_{i=j}^{n} \psi_j(o_j, \pi_{o_j}) \\
&= \operatorname*{argmax}_{O^i, i=1,\ldots,m} \max_{X \in H} p'(O^i, H)
\end{aligned}
$$

so that, once again, our ranking of each $O^i$ based on Viterbi score does not depend on the normalizing constants $Z_1, \ldots, Z_n$.

## 1.3 Viterbi path

Consider we'd like to compute the Viterbi path $H^*$ of $G$ given $O$. We thus have

$$
\begin{aligned}
H^* &= \operatorname*{argmax}_{X \in H} \frac{1}{Z} \mathrm{p}(H|O, \pi_O) \prod_{i=1}^n \psi_i(o_i, \pi_{o_i}) \\
&= \operatorname*{argmax}_{X \in H} \mathrm{p}(H|O, \pi_O) \prod_{i=1}^n \psi_i(o_i, \pi_{o_i}) \\
&= \operatorname*{argmax}_{X \in H} \mathrm{p}'(O, H).
\end{aligned}
$$

Thus, computing the Viterbi path does not require the normalizing constants $Z_1, \ldots, Z_n$.

## 2 Proofs for Convex Virtual Emissions

Assume that we're given a Bayesian network where $E$ is the set of observed evidential random variables, $H$ is the hypothesis space composed of the cross-product of the $n$ hidden discrete random variables in the network, and $h \in H$ is an arbitrary *hypothesis* (i.e., an arbitrary instantiation of the hidden variables). We assume the observed variables $e \in E$ are leaf nodes, as is common in many time-series models where the hidden layer explains the downstream observed layer and and observed nodes do not share edges, such as hidden Markov models (HMMs), hierarchical HMMs [7], dynamic Bayesian networks (DBNs) for speech recognition [2], hybrid HMMs [1], as well as the DBNs designed for MS/MS analysis, DRIP [3] and Didea [9].

Consider the log-posterior probability

$$
\log p(h|E) = \log \frac{p(h, E)}{p(E)} = \log \frac{p(h, E)}{\sum_{\bar{h} \in H} p(\bar{h}, E)} = \log p(h, E) - \log \sum_{h \in H} p(h)p(E|h), \tag{2}
$$

as is computed in Didea. Assume $p(h)$ and $p(E|h)$ are non-negative probability distributions and that the emission density $p(E|h)$ is parameterized by $\theta$ which we'd like to learn. Applying virtual evidence for such models, $p(E|h)$ need not be normalized for posterior inference, Viterbi inference, and comparative inference between sets of observations (further details are discussed in Section 1).

Assume that there is a parameter $\theta_h$ to be learned for every hypothesis of latent variables, though if we have fewer parameters, parameter estimation becomes strictly easier. We make this parameterization explicit by denoting the emission distributions of interest as $p_{\theta_h}(E|h)$. We first look for functions $f_h(\theta_h) = p_{\theta_h}(E|h)$ which render the log-likelihood convex,

$$
\log p(E) = \log \sum_{h \in H} p(h)p_{\theta_h}(E|h) = \log \sum_{h \in H} c_h f_h(\theta_h), \tag{3}
$$

where $c_h = p(h)$ are nonnegative constants with regards to the parameters of interest. Assume that $f_h(\cdot)$ is smooth on $\mathbb{R}$ for all $h \in H$.

**Theorem 1.** *The convex functions of the form* $\log \sum_{h \in H} c_h f_h(\theta_h)$, *such that* $(f_h'(\theta_h))^2 - f_h''(\theta_h)f_h(\theta_h) = 0$, *are* $\log \sum_{h \in H} c_h f_h(\theta_h) = \log \sum_{h \in H} c_h \alpha_h e^{\beta_h \theta_h}$, *where* $\alpha_h$ *and* $\beta_h$ *are constants uniquely determined by initial conditions.*

*Proof.* In order to ensure convexity of Equation 3, it is necessary and sufficient that $\nabla_\theta^2 \log \sum_{h \in H} c_h f_h(\theta_h) \succeq 0$. We thus have the following for the gradient

$$
\nabla_\theta \log \sum_{h \in H} c_h f_h(\theta_h) = \frac{1}{\sum_{h \in H} c_h f_h(\theta_h)} \begin{bmatrix} c_1 f_1'(\theta_1) \\ \vdots \\ c_{|H|} f_{|H|}'(\theta_{|H|}) \end{bmatrix}.
$$

Letting $Z = \sum_h c_h f_h(\theta_h)$, we have

$$
\frac{\delta \log p_\theta(h|E)}{\delta \theta_i \delta \theta_j} = \begin{cases} \frac{c_i f_i''(\theta_i)Z - (c_i f_i'(\theta_i))^2}{Z^2} & \text{if } i = j \\ \frac{-c_i f_i'(\theta_i)c_j f_j'(\theta_j)}{Z^2} & \text{if } i \neq j \end{cases}
$$

Letting $a = \begin{bmatrix} c_1 f_1''(\theta_1) & \cdots & c_{|H|} f_{|H|}''(\theta_{|H|}) \end{bmatrix}^T$ and $b = \begin{bmatrix} c_1 f_1'(\theta_1) & \cdots & c_{|H|} f_{|H|}'(\theta_{|H|}) \end{bmatrix}^T$, we may thus write the Hessian as

$$\nabla_\theta^2 \log \sum_{h \in H} c_h f_h(\theta_h) = \frac{\text{diag}(a)}{Z} - \frac{1}{Z^2} bb^T. \tag{4}$$

Equation 4 is positive semi-definite if and only if, for all $x \in \mathbb{R}^n$,

$$x^T \nabla_\theta^2 \log \sum_{h \in H} c_h f_h(\theta_h) x \geq 0$$

$$Z^2 x^T \nabla_\theta^2 \log \sum_{h \in H} c_h f_h(\theta_h) x \geq 0$$

$$x^T (Z \text{diag}(a) - bb^T) x \geq 0$$

$$x^T Z \text{diag}(a) x \geq x^T bb^T x$$

$$\left( \sum_i c_i f_i(\theta_i) \right) \left( \sum_i c_i f_i''(\theta_i) x_i^2 \right) \geq \left( \sum_i c_i f_i'(\theta_i) x_i \right)^2. \tag{5}$$

Letting $l, u, v$ be vectors with components $l_i = x_i \sqrt{c_i f_i''(\theta_i)}$, $u_i = x_i \frac{c_i f_i'(\theta_i)}{\sqrt{c_i f_i(\theta_i)}}$, $v_i = \sqrt{c_i f_i(\theta_i)}$, we require

$$(l^T l)(v^T v) \geq (u^T v)^2. \tag{6}$$

Note that, by the non-negativity of $c_h$ and $f_h(\theta_h)$, $v_i$ is real and the quantify $l^T l$ is always real. When $l = u$, the bound in Equation 6 is guaranteed to hold by the Cauchy-Schwarz inequality. Thus, the log-probability of evidence is convex when

$$l_i = u_i$$

$$\Rightarrow x_i \sqrt{c_i f_i''(\theta_i)} = x_i \frac{c_i f_i'(\theta_i)}{\sqrt{c_i f_i(\theta_i)}}$$

$$\Rightarrow f_i''(\theta_i) f_i(\theta_i) = (f_i'(\theta_i))^2. \tag{7}$$

Equation 7 is an autonomous, second-order, nonlinear ordinary differential equation (ODE), the solution of which leads us to the following generalization of the commonly encountered LSE convex function.

To simplify notation, let $t = \theta_i$ and $y(t) = f_i(t)$, where we drop the independent variable when it is understood. Thus, we are looking for $y$ such that

$$y'' = \frac{(y')^2}{y}. \tag{8}$$

Let $v(t) = y'(t)$, $w(y) = v(t(y))$, and $\dot{w} = \frac{dw}{dy}$. We thus have

$$v' = \frac{v^2}{y},$$

$$\dot{w}(y) = \frac{dw}{dy}(y) = \frac{dv}{dt} \frac{dt}{dy} \Big|_{t(y)} = \frac{v'}{y'} \Big|_{t(y)} = \frac{v'}{v} \Big|_{t(y)}.$$

Using Equation 8, we have

$$\dot{w}(y) = \frac{v'(t(y))}{w(y)} = \frac{v^2(t(y))}{w(y)y} = \frac{w^2(y)}{w(y)y} = \frac{w(y)}{y}.$$

Solving this ODE using seperation of variables, we have

$$\ln w = \ln y + d_0$$

$$\Rightarrow w = \exp(\ln y + d_0) = e^{d_0} y = d_1 y,$$

where $d_0$ is a constant of integration. To solve for $y$, we have
$$w(y(t)) = v(t) = y' = d_1 y.$$
As before, we solve $y' = d_1 y$ using seperation of variables, giving us
$$\ln y = d_1 t + d_2$$
$$\Rightarrow y(t) = d_3 e^{d_1 t},$$
where $d_1$ and $d_3$ are constants uniquely determined by initial conditions. Returning to our earlier notation, the solution to Equation 7 is thus $f_i(\theta_i) = d_3 e^{d_1 \theta_i}$. Letting $\beta_i = d_1$ and $\alpha_i = d_3$ completes the proof. $\square$

**Corollary 1.1.** *For convex* $\log p(E) = \log \sum_{h \in H} p(h) p_{\theta_h}(E|h)$ *such that* $(p'_{\theta_h}(E|h))^2 - p''_{\theta_h}(E|h) p_{\theta_h}(E|h) = 0$, *the log-posterior* $\log p_\theta(h|E)$ *is concave in* $\theta$.

*Proof.* From Theorem 1, $p_{\theta_h}(E|h) = \alpha_h e^{\beta_h \theta_h}$ and we have
$$\log p_\theta(h|E) = \log p_{\theta_h}(h, E) - \log \sum_{h \in H} p(h) p_{\theta_h}(E|h)$$
$$= \log p(h) p_{\theta_h}(E|h) - \log \sum_{h \in H} p(h) p_{\theta_h}(E|h)$$
$$= \log p(h) \alpha_h e^{\beta_h \theta_h} - \log \sum_{h \in H} p(h) p_{\theta_h}(E|h)$$
$$= \log p(h) \alpha_h + \beta_h \theta_h - \log \sum_{h \in H} p(h) p_{\theta_h}(E|h).$$
With respect to $\theta$, $\beta_h \theta_h$ is affine, $-\log \sum_{h \in H} p(h) p_{\theta_h}(E|h)$ is concave, and the remaining term is constant. $\square$

# 3 Analysis of CVEs in Didea

In order to analyze Didea's scoring function, define boolean vectors length $\bar{o}$ $b_x, y_x$ such that, for the set of b-ions $\beta_x = \{\cup_{i=1}^{l-1} \{b(m(x_{1:i}), 1)\}\}$ and y-ions $v_x = \{\cup_{i=0}^{l-1} \{y(m(x_{i+1:l}), 1)\}\}$ of $x$, and $0 \le j \le \bar{o}$ we have
$$b_x(j) = \mathbf{1}_{\{j \in \beta\}}, \quad y_x(j) = \mathbf{1}_{\{j \in v\}}.$$
We note that computing Didea scores as detailed in the sequel would be much more slower than computing Didea scores using sum-product inference. However, the compact description of Didea's scoring function allows much easier analysis in Sections 3.1 and 3.2.

Recall the CVE used in the main paper, $f_{\theta_\tau}(s(i)) = e^{\theta_\tau s(i)}$. Under this new emission distribution, Didea's scoring function may thus be compactly written as
$$\psi_\lambda(s, x) = \log \mathrm{p}(x, z|\tau_0 = 0) - \log \sum_\tau \mathrm{p}(x, z_\tau|\tau_0 = \tau)$$

$$= \sum_{t=1}^{l} (\log f_{\theta_0}(b_t) + \log f_{\theta_0}(y_{n-t})) - \log \sum_\tau \exp \sum_{t=1}^{l} (\log f_{\theta_\tau}(b_t) + \log f_{\theta_\tau}(y_{n-t}))$$
$$= \theta_0 b_x^T s + \theta_0 y_x^T s - \log \sum_\tau \exp(\theta_\tau b_x^T s_\tau + \theta_\tau y_x^T s_\tau) = \theta_0 (b_x + y_x)^T s - \log \sum_\tau \exp\left[\theta_\tau (b_x + y_x)^T s_\tau\right].$$
$$(9)$$

## 3.1 Gradients of CVEs in Didea

Letting $h_\tau(x, s) = (b_x + y_x)^T s_\tau$, the gradient of this new conditional log-likelihood has elements
$$\frac{\delta}{\delta \theta_\tau} \psi_\theta(s, x) \bigg|_{\tau=0} = h_0(x, s) - \frac{1}{\sum_0 e^{\theta_0 h_0(x,s)}} \sum_0 h_0(x, s) e^{\theta_0 h_0(x,s)} \tag{10}$$

$$\frac{\delta}{\delta \theta_\tau} \psi_\theta(s, x) \bigg|_{\tau \neq 0} = -\frac{1}{\sum_\tau e^{\theta_\tau h_\tau(x,s)}} \sum_\tau h_\tau(x, s) e^{\theta_\tau h_\tau(x,s)}. \tag{11}$$

Given $N$ i.i.d. training PSMs $\{(s^1, x^1), (s^2, x^2), \dots, (s^N, x^N)\}$, we need only run sum-product inference once to cache the values $\{\cup_{i=1}^N \{h_{-L}(s^i, x^i), \dots, h_0(s^i, x^i), \dots, h_L(s^i, x^i)\}\}$ for extremely fast gradient based learning.

## 3.2 Proof of Didea lower bound for the XCorr scoring function

**Theorem 2.** *Assume the PSM scoring function $\psi(s, x)$ is that of Didea under the emission function $f_{\theta_\tau}(s(i))$ with uniform weights $\theta_i = \theta_j$, for $i, j \in [-L, L]$. Then $\psi(s, x) \leq \mathcal{O}(XCorr(s, x))$.*

*Proof.* Recall that, for theoretical spectrum $u$, XCorr is computed as

$$\text{XCorr}(s, x) = u^T s - \frac{1}{2L+1} \sum_{\tau=-L}^{L} u^T s_\tau = u^T (s - \frac{1}{2L+1} \sum_{\tau=-L}^{L} s_\tau) = u^T s'.$$

Let $\lambda = \theta_i$ for $i \in [-L, L]$. From Didea's scoring function, we have

$$\psi(s, x) = \log p(x, z, \tau_0 = 0) - \log \sum_{\tau=-L}^{L} p(\tau_0 = \tau)p(x, z_\tau | \tau_0 = \tau) = \log p(x, z, \tau_0 = 0) - \log \mathbf{E}[p(x, z_\tau | \tau)],$$

so that, by Jensen's inequality,

$$\psi(s, x) \leq \log p(x, z, \tau_0 = 0) - \mathbf{E}[\log p(x, z_\tau | \tau)]. \tag{12}$$

The right-hand side of 12, which we'll denote as $g(s, x)$, is

$$g(s, x) = \log \frac{1}{|\tau|} p(x, z | \tau_0 = 0) - \mathbf{E}[\log p(x, z_\tau | \tau_0 = \tau)]$$

$$= -\log |\tau| + \lambda(b_x + y_x)^T s - \sum_{\tau=-L}^{L} p(\tau_0 = 0) \log e^{\lambda(b_x + y_x)^T s_\tau}$$

$$= -\log |\tau| + \lambda(b_x + y_x)^T s - \frac{\lambda}{|\tau|} \sum_{\tau=-L}^{L} (b_x + y_x)^T s_\tau.$$

Letting $u = b_x + y_x$, we have

$$g(s, x) = -\log |\tau| + \lambda u^T s - \frac{\lambda}{|\tau|} \sum_{\tau=-L}^{L} u^T s_\tau = -\log |\tau| + \lambda u^T (s - \frac{1}{|\tau|} \sum_{\tau=-L}^{L} s_\tau) = -\log |\tau| + \lambda \text{XCorr}(s, x).$$

$$\Rightarrow \psi(s, x) \leq g(s, x) = -\log |\tau| + \lambda \text{XCorr}(s, x) \qquad \square$$

## 4 Charge varying spectra

In practice, observed spectra exhibit higher charged fragment ions. In order to account for these new fragmentation peaks while still keeping the scoring function well calibrated (i.e., keeping higher charged PSMs comparable in range to lower charged PSMs), a Didea charge varying model is introduced in [9]. In this model, a global variable switches between two separate models: the singly charged model and a model which considers both single and double charged fragment ions. The latter model contains a charge variable in every frame which is hidden and integrated over. This effectively averages the contribution between the differently charge b-and y-ion pairs. Finally, the contribution between the two separate charged models is averaged (per frame). The posterior of $\tau_0 = 0$ remains Didea's PSM score in this setting. Further details of the model's scoring function may be found in [9].

## 5 Conditional Fisher kernel for improved discriminative analysis

We leverage Didea's gradient-based PSM information to aid in discriminative postprocessing analysis. We utilize the same set of features as the DRIP Fisher kernel [5] where, to measure the relative utility of the gradients under study, the DRIP log-likelihood gradients are replaced with Didea gradient information (derived in Section 3.1). These features are used to train an SVM classifier, Percolator [6], which recalibrates PSM scores based on the learned decision boundary between input targets and

decoys. The resulting Didea conditional Fisher kernel is benchmarked against the DRIP Fisher kernel and the scoring algorithms benchmarked in the main paper (using their respective standard Percolator features sets). DRIP Kernel features were computed using a customized version of the DRIP Toolkit, provided by the authors of [5]. MS-GF+ Percolator features were collected using `msgf2pin` and XCorr/XCorr $p$-value features collected using Crux. The resulting postprocessing results are displayed in Figure 1.

Figure 1: Post-database-search accuracy plots measured by $q$-value versus number of spectra identified for worm (*C. elegans*) and yeast (*Saccharomyces cerevisiae*) datasets. All methods are post-processed using the Percolator SVM classifier [6]. "DRIP Fisher" augments the standard set of DRIP PSM features (described in [4]) with the recently derived gradient-based DRIP features in [5]. "Didea Fisher" uses the aforementioned DRIP features with the gradient features replaced by Didea's conditional log-likelihood gradients. "XCorr," "XCorr $p$-value," and "MS-GF+" use their standard sets of Percolator features (described in [4]).