[Reviews · NeurIPS 2018]

Reviewer 1



The Authors address the task of matching peptides (protein fragments) to tandem MS (mass spectrometry) spectra, an important step in identifying the protein content of biological samples, and expand the parameter learning capabilities of Didea, a dynamic Bayesian network (DBN) scoring function, by deriving Convex Virtual Emissions (CVEs), a class of emission distributions for which conditional maximum likelihood learning is concave. CVEs are shown to generalize the widely-used log-sum-exp function; and incorporating CVEs into Didea greatly improves its PSM identification accuracy. The paper is clearly written, though I’d argue for a different split of topics between the main text and supplementary information. To the best of my knowledge, CVEs are novel and of wide applicability; and, importantly, the demonstrated improvements in accuracy and run time seem convincing. Specific comments 1. Section 4.3 (and the corresponding supplementary text) proves that Didea’s scoring function lower bounds XCorr’s. While an interesting result and, as noted, a promising future direction, it seems somewhat orthogonal to the main theme of the paper; perhaps move section 4.3 to supplementary information or save all discussion for a separate future publication? 2. Reported improvement in performance in sections 5 (multi-parameter Didea) and 5.1 (+ Fisher kernel post-processing) are numerically identical; is this expected? A coincidence? 3. Fig 4, why not combine with Fig 3, and show all methods side-by-side? Minor comments Line 121: (described Section 4.3), “in” is missing; l158 and Fig 2: equation for M’_t seems wrong, should be M’_{t+1}? l171: log p(\tau_0, x, z), should be s instead of z?

Reviewer 2



The manuscript “Learning Concave Conditional Likelihood Models for Improved Analysis of Tandem Mass Spectra” extends a dynamic Bayesian network approach called DIDEA by introducing a new class of emission distributions. The conditional log-likelihood of those functions remains concave leading to an efficient global optimization method for parameter estimation. This is in stark contrast to the previous variant, for which the best parameter had to be found by grid search. In comparison to other state-of-the-art methods, the new approach outperforms the other methods, while being faster at the same time. Quality Overall the quality of the manuscript is good. There were some errors in the formulas, but the authors promised to fix them at the author response period. Clarity Overall the clarity is good, there are just some parts, where the reader has to guess certain details (e.g., regarding b and y). It is a little bit confusing that in the introduction it is mentioned that DRIP outperformed DIDEA (line 39), but then in the comparison on the data sets where the authors show the performance of the old DIDEA approach (Figure 4, Didea-0), the Didea-0 performance seems to be superior to the performance of DRIP (shown in Figure 3). It would have been better to show Didea-0 also in Figure 3. Showing this separately does not make a good impression especially in light of the above mentioned observation. Figure 4 also seems to imply that the performance gain is not as substantial as claimed in line 281. Author response: ...Please see the introductory DRIP paper [1] being referred to on line 39, where the authors show that DRIP outperforms Didea... We note, though, that the authors of [1] only test algorithms by assuming a fixed observed precursor charge of 2, rather than the harder charge-varying case considered in this paper. answer: But then you are saying now that in the charge-varying case DRIP is worse than the old DIDEA. One should at least discuss this and not just copy an old statement from a paper with a non-extensive evaluation. It is good to see a statement about the runtime in the manuscript (with actual values) and not just in the supplement. This seems to be the main selling point in case the performance of Didea and Didea-0 is really that close. Significance The introduced distributions, called convex virtual emissions (CVEs), seem interesting and might have applicability beyond DBNs, but the focus of the manuscript is only on this one application of DBNs with CVEs for Tandem Mass Spectra analysis. If the authors could make plausible that CVEs could be useful for other models, then this would be of even higher interest to the general NIPS community.

Reviewer 3



SUMMARY This work is about the identification of peptide sequences in tandem mass spectrometry. The underlying problem is to assemble a set of mass peaks representing peptide fragments to a full peptide sequence (which would be the de-novo version of this problem), or to derive a matching score for petptides in a database (which is the version considered here). The de-facto standard methods for this purpose are variants of HMMs or dynamic Bayes nets. The main idea proposed here is to use a specific emission distribution which leads to a concave (conditional) likelihood, thereby simplifying and speeding-up inference. EVALUATION: clarity: the paper is easy to read, and together with the supplementary material, the derivation is easy to follow. novelty & significance: the main methodological contribution is the definition and characterization of Bayesian network emission distributions which generalize the convex log-sum-exp function. Applied in the contex of DBN scoring within the Didea model, the conditional log-likelihood becomes concave in the parameters, thus giving theoretical gurantees for convergence to the global optimum. I consider this as a significant contribution. experimental evaluation: equipped with the new emission models, Didea has been shown to outperform state-of-the-art scoring algorithms on bench-mark datasets.